# Effect of Grain Size on the Tribological Behavior of CoCrFeMnNi High Entropy Alloy

**DOI:** 10.3390/ma16041714

**Published:** 2023-02-18

**Authors:** Ying Wang, Dong Li, Junsheng Yang, Junsong Jin, Mao Zhang, Xinyun Wang, Bin Li, Zhigang Hu, Pan Gong

**Affiliations:** 1School of Mechanical Engineering, Wuhan Polytechnic University, Wuhan 430000, China; 2State Key Laboratory of Materials Processing and Die & Mould Technology, School of Materials Science and Engineering, Huazhong University of Science and Technology, 1037 Luoyu Road, Wuhan 430074, China

**Keywords:** high entropy alloy, grain size, tribological behavior, wear mechanism

## Abstract

The effect and mechanism of grain sizes on the tribological behavior of CoCrFeMnNi high entropy alloy (HEA) were studied by friction experiments and wear morphology analysis. Under normal low load and low sliding speed, the primary wear mechanism of the HEA samples is adhesive wear. With the increase in sliding speed, the wear mechanisms of the samples are adhesive wear and oxidation wear. The oxide layer formed under the action of friction heat of the coarse grain (CG) sample is easy to break due to the softening of the CG. With the increase of normal load and sliding speed, the wear mechanisms of the HEA samples are mainly adhesive wear, oxidation wear, and plastic deformation. The oxide layer of CG sample has many cracks, and the worn surface also has plastic deformation, which leads to the increase of friction coefficient and specific wear rate and the decrease of wear resistance. Therefore, the fine grain size HEA sample has better wear resistance than the CG sample due to its high surface strength.

## 1. Introduction

High entropy alloys (HEAs) are different from traditional alloys, which are composed of multiple principal elements. Due to the high entropy effect, slow diffusion effect, lattice distortion, and cocktail effect [1], HEAs have unique and excellent characteristics, such as high strength [2,3], excellent oxidation resistance at elevated temperature [4,5], and good wear resistance [6,7], which have received extensive attention from domestic and foreign scholars. CoCrFeMnNi HEA shows excellent mechanical properties due to the forming a single FCC phase, with excellent elongation of 60–70%, tensile strength of more than 1 GPa, fracture toughness of more than 200 GPa, and excellent damage resistance [8]. Therefore, CoCrFeMnNi HEA is an essential candidate for new materials of critical components in industrial applications, such as die manufacturing, aerospace, defense-heavy industry, and nuclear industry [9,10,11,12]. In these practical industrial applications, friction and wear behavior is essential. These key components and parts need well mechanical properties and outstanding wear resistance [13,14,15]. The moving parts need excellent friction pair materials to extend their service life. Therefore, the wear resistance of the CoCrFeMnNi HEA is a vital evaluation index before industrial application.

The microstructure greatly influences the friction behavior of HEAs [16]. The addition of different elements in CoCrFeMnNi HEAs changes their microstructure and thus affects their wear resistance [17]. The addition of different amounts of Nb in Al_0.25_CrFeNi_1.75_Nb_x_ HEA results in the formation of AlCrNi type and AlN_3_ type, which exhibit excellent hardness and wear resistance and effectively improve the friction and wear behaviors of the HEA [18]. The doping of Si into Al_0.2_Co_1.5_CrFeNi_1.5_Ti_0.5+0.5x_Si_x_ (x = 0.1, 0.3 and 0.5) HEA effectively promotes the formation of Ti_5_Si_3_ and TiSi [19]. The second phase of these solid solutes and silicide effectively improves the hardness of the HEA. The friction properties of the HEA can be improved effectively with the increase of the content of Si. The (FeMnNiCoCr)N_x_ coatings with different N contents were prepared by magnetron sputtering. It was found that the coatings with medium nitrogen (~6 at. %) and high nitrogen (~26 at. %) coatings of BCC structures have high hardness and wear resistance [20].

In addition to changing the element composition, changing the preparation method can also improve the wear resistance of the HEA. CoCrFeMnNi HEA prepared by powder metallurgy has grain boundary and dislocation strengthening with increasing grinding time. The hardness of the HEA increases from 270 ± 10 Hv to 450 ± 10 Hv, and the wear resistance of the HEA is effectively improved [21]. At present, many scholars also apply various techniques to prepare HEA coatings, such as magnetron sputtering, laser cladding, plasma cladding, spark deposition, and thermal spraying [22]. The microstructure of the HEA coatings has a decisive effect on the wear properties. The wear rate gradually decreases with the increase of Al content in the FeCoNiCrSiAl_x_ HEA coatings due to the formation of different phase structures with different Al content. The lowest wear rate of FeCoNiCrSiAl_1.0_ HEA coating results from the formation of BCC and Cr_3_Ni_5_Si_2_ phases [23]. Changing the microstructure of the CoCrFeMnNi HEA by different methods has a significant effect on its wear resistance.

In addition to changing the composition of elements and the preparation method, the microstructure of the CoCrFeMnNi HEA can be changed by different heat treatment processes. However, research on the wear behavior of CoCrFeMnNi HEA obtained by different heat treatment processes needs to be completed. Different grain sizes of CoCrFeMnNi HEA are obtained after different heat treatment processes. In order to better apply CoCrFeMnNi HEA to industrial manufacturing and consummate the study of the influence of microstructure on the wear behavior of CoCrFeMnNi HEA, it is necessary to explore the wear resistance of annealed CoCrFeMnNi HEAs. Based on the above considerations, the influence and mechanism of grain size on friction and wear behaviors of CoCrFeMnNi HEA under different loads and speeds are discussed systematically using nanoindentation, friction test, scanning electron microscope, and a white light interference diffractometer.

## 2. Materials and Methods

### 2.1. Experimental Materials

Alloy ingots with a nominal composition of CoCrFeMnNi HEA were synthesized by arc-melting a mixture of corresponding commercial pure metals (purity > 99 wt. %) [24]. The samples were re-melted three times, making the elements distributed homogeneously. The melted alloys were eventually drop-cast into a copper mold. The final cylindrical ingots with 110 mm diameter and 90 mm height were obtained. The composition is shown in Table 1. In order to reduce the internal defects of castings, the ingots were forged at 1000 °C, and then annealed at 1000 °C for 3 h. The chemical composition of CoCrFeMnNi HEA can be seen in Table 1, which is measured using energy dispersive X-ray (EPMA-8050G, Shimadzu, Shanghai, China). The slab sample with a section size of 50 mm (length) × 10 mm (width) × 5 mm (thickness) was cut by a wire cutting machine. At room temperature, the slab was rolled through multi-pass cold rolling. The different grain size samples were attained by different annealing temperatures. The samples thinned 80% by cold rolling were held at 800 °C for 30 min and 1000 °C for 60 min, respectively, to obtain the fine grain sample (FG) and coarse grain sample (CG). The sliding wear test sample with a section size of 20 mm × 10 mm was processed by a wire-cutting machine from the cold rolling samples. The surfaces of all the sliding wear test samples were ground with #400-2000 silicon carbide sandpapers and polished to a mirror surface with alumina suspension to ensure good roughness and surface quality.

### 2.2. Sliding Fiction Test

The tribological behavior of the samples was evaluated by reciprocating sliding friction and wear tests using a ball-type friction tester (UMT trilobal machine, Bruker, Saarbrucken, Germany). The CoCrFeMnNi HEA sample was rubbed against a hardened Si3N4 ball with a diameter of 10 mm. The hardened Si3N4 ball has excellent wear resistance, does not react with the alloy, and will not adhere to the sample surface, which ensures the stability of the friction test. All the wear tests were dry friction under different loads and sliding speeds at room temperature. The normal load of the friction test is 5 N and 30 N, the sliding speed is 1 mm/s, 5 mm/s, and 10 mm/s, the friction length is 5 mm, and the sliding time is 1200 s. The balls and samples should be cleaned with acetone ultrasonic for 20 min before each friction and wear test. In order to ensure the validity and repeatability of friction test data, each friction coefficient curve was obtained through three reciprocating wear experiments under the same condition. The friction coefficient (µ) was continuously recorded by the friction tester during the whole wear test. The wear volume after the wear test is obtained by the three-dimensional white light interferometer (VHX-6000, Keyence, Osaka, Japan) to measure the depth and width of the wear track. Different areas of the friction track were selected for testing each time, and the width and depth of the track were measured 5 times to ensure repeatability. The schematic diagram of a typical adhesive track and white light interference 3D contour image is shown in Figure 1. The specific wear rate was calculated according to Archard’s law [25], and the equation is as follows:(1)$$K=V/\left(F\cdot L\right)$$
where *V* is the wear volume, *F* is the normal force, and *L* is the total sliding distance. The specific wear rate and sliding friction coefficient (µ) of the FG and CG samples are the mean values of the three fiction and wear tests.

### 2.3. Microstructure Characterization of the Friction Sample

The microstructural features of different grain size samples were characterized by using Quanta 650FEG scanning electron microscopy (SEM) equipped with an electron backscatter diffraction (EBSD) and an energy dispersive X-ray spectrometer (EDS) detector. The surface morphology and composition of the wear tracks were analyzed using SEM and EDS. The hardness of the different grain size samples was measured using the Nano track tester (TI750, Bruker) with the Berkovich triangular cone indenter.

## 3. Results and Discussion

### 3.1. Microstructure Characterization of CoCrFeMnNi HEA with Different Grain Size

The microstructure of the sample was characterized by EBSD, and the corresponding inverse pole diagram (IPF) and corresponding grain size distribution map were obtained by OIM analysis software, as shown in Figure 2. The step size of EBSD tests was 2.5 μm. The neighbor CI correlation method was used to clean the EBSD data. The diameter of grain size was determined by grain area. It can be seen in that a large number of equiaxed crystals and annealing twins are formed in the sample after heat treatment. The grain size after recrystallization increases with annealing temperature and annealing time. The samples with different grain sizes can be obtained by heat treatment at different temperatures and time of cold-rolled HEAs. The grain size of the sample was quantitatively obtained using the analysis software OIM in EBSD. The average grain size of the sample is 2.9 μm, defined as fine grain (FG), shown in Figure 2a. The average grain size of the sample is 63.4 μm, defined as coarse grain (CG), shown in Figure 2b.

The load and displacement curve of the sample measured by the nanoindentation test is shown in Figure 3. It can be seen that the hardness of FG sample is significantly greater than that of CG sample, and the hardness of FG and CG sample is approximately 4.9 GPa and 3.4 GPa, respectively. The results show that the hardness and strength of HEAs can be significantly improved by refining grains, increasing the resistance of slip band and dislocation movement, which has been confirmed by relevant literature [26,27].

### 3.2. Friction Coefficient

The friction behavior of HEA under dry friction conditions is researched in this section in terms of studying the effect of normal load and sliding velocity on the friction coefficient. Figure 4 shows the friction coefficient curves of FG and CG samples under different normal loads and sliding friction speeds. It is evident from Figure 4 that the friction coefficient in the high loads is lower than that of the low loads, which is consistent with the metallic alloys in dry sliding [28]. All the friction coefficient curves rise rapidly, reach a stable value, and then fluctuate around the stable value. Generally, there are some rough structures with low hardness on the surface of the alloy, which would gradually break and fill into the surrounding grooves during dry sliding. The curve is relatively smooth at low speed and low load condition, while the rough materials struggle to fill the groove with the increase of speed and load, so the friction coefficient curve fluctuates greatly [29]. Under a low load of 5 N and a sliding speed of 1 mm/s, the curve trends of the FG and CG samples are close and smooth. When the sliding speed increased to 5 mm/s, the curves trend of the two samples is also similar, and the friction coefficient of CG is greater than that of FG. It can be seen from Figure 4d that the friction coefficient curve of FG sample fluctuates sharply with the increase of normal force. This is because the oxidation reaction occurred suddenly at the interface of some parts in the wear track with the increase of normal force, forming discontinuous oxides, which causes drastic fluctuations. This corresponds to the analysis of corresponding wear track elements. When the sliding speed is 10 mm/s, the friction coefficient curve of CG fluctuates dramatically while that of FG also has some small fluctuations. In Figure 4e, when the sliding speed increases, with the increase of sliding time, the oxidation reactions also occur in some parts of the wear track, resulting in sharp fluctuations in the friction coefficient curves of the FG sample. This is due to the increase of friction rate and the formation of a large number of grinding chips on the surface, resulting in a large degree of the concave-convex surface. Under the load of 30 N, the fluctuation degree of the FG and CG samples is more severe than that of the low load of 5 N. As shown in Figure 4f, with the increase of normal load and sliding speed, the oxidation reaction occurs on the sliding surface due to the friction heat, producing oxides and even plastic deformation on the track surface, resulting in curve fluctuations. However, a large number of oxides form a continuous oxide layer at the later friction stage. The oxide layers play a role in lubrication at the later stage of friction. The fluctuation of the curve decreases gradually, and the friction coefficient of the FG sample decreases. Therefore, increasing the normal load and sliding friction speed will cause significant fluctuations in the friction coefficient curve due to the accumulation and elimination of repeated grinding on the sample surface during sliding.

Figure 5 shows the average friction coefficients of FG and CG samples under low and high loads and different sliding speeds. The results indicate that the friction coefficient under the high load is lower than that under low load. As the sliding friction speed increases, the friction coefficient increases. However, increasing the normal load will lead to a decrease in the friction coefficient. Friction behavior can be determined by the chemical reaction kinetics between friction pairs, which is mainly caused by friction energy and strain energy. When the normal load increases, the overheating friction heat at the friction interface causes the HEA to oxidize in the air atmosphere and form the oxide film. The oxide film can play a role in lubrication and reduce the shear strength of the friction interface, thus inhibiting the plastic deformation caused by repeated wear of the surface and ultimately reducing the friction coefficient. Therefore, with the increase of normal load, more friction heat is generated, which makes the friction surface smooth and decreases the friction coefficient. This is due to the high hardness and surface strength of FG sample. The friction coefficient of FG sample is more minor than the CG sample under the same sliding speed and load conditions.

### 3.3. Specific Wear Rate Analysis

In order to better analyze the wear behavior of FG and CG samples under different normal loads and sliding speeds, the wear track was analyzed using a three-dimensional white light interferometer. The cross-sectional profiles of the wear tracks are presented in Figure 6. The curves are not smooth, caused by the constant accumulation and elimination of wear debris and the breakage of the wear surface, resulting in uneven wear areas. Under low loading conditions, the cross-sections of FG and CG samples are very similar, and the cross-section sizes increase slowly with the increase of sliding speed. Compared with this, under the high load condition, the section profile of the two samples increases obviously. At the same time, it can be seen that under low sliding speed, the cross-section profile of CG sample is much larger than that of FG sample. As the sliding speed increases to 5 mm/s and 10 mm/s, the section profiles of the two samples both increase, but the gap between the two samples decreases. The section width of the FG and CG samples tends to be the same, but the depth of the CG sample is much larger than that of FG sample.

Figure 7 and Figure 8 show the surface morphology of the FG and CG samples observed by the 3D optical microscope in the normal load of 5 N and 30 N under different sliding speeds, respectively. The color change in the figure indicates the depth change of the wear track. The wear tracks of FG and CG samples under the normal load of 5 N are more significant than that under the normal load of 30 N. Under the same normal load with different sliding speeds, the cross-section width of tracks in samples presents little difference. Through careful consideration, the track width of the two samples is the largest under the normal load of 30 N and the speed of 10 mm/s. At a high load of 30 N, there are some black regions on the track. The element analysis shows that these areas are rich in O elements. It is inferred that the oxidation reaction occurred due to friction heat.

The specific wear rate of samples with different grain sizes is displayed in Figure 9. The specific wear rate of the FG and CG samples after sliding was calculated by using Equation (1). As can be seen from the figure, with the increase of normal load, the wear degree of the alloy surface will be increased, and the specific wear rate will decrease. As the sliding speed increases, the wear degree of the alloy surface will be increased, resulting in the increase of specific wear rate in both samples. In addition, it can be obviously observed that the specific wear rate of FG sample is significantly lower than that of the CG sample, indicating that the sample obtained by refining grain has better wear resistance. The fine-grained sample has higher surface hardness and mechanical strength, which is helpful to improve the wear resistance [30]. Compared with other materials, this can give us some inspiration for later work. The average wear rate of TiO_2_ 10 wt.% NiAl coating deposited on the low alloy steel 40Cr4 surface is 9.821 × 10^−5^ mm^3^ N^−1^ m^−1^, and the wear rate of TiO_2_13 wt.% NiAl coating 40Cr4 is 1.22 × 10^−5^ mm^3^ N^−1^ m^−1^ under the normal load of 10 N and sliding speed of 500 mm/s [31], which is close to that of FG HEA under the load of 30 N and sliding speed of 5 mm/s. The wear rate of AlCrSiN-coated tool steel K340 is 5.681 × 10^−7^ mm^3^ N^−1^ m^−1^ under the load of 10 N and 10 cm/s [32]. The average wear rate of AlTiSiN + TiSiN coatings deposited on the H 13 tool steels is 2.30 × 10^−8^ and 1.31 × 10^−8^ mm^3^ N^−1^ m^−1^ for normal load of 30 and 50 N at the lubricated conditions, respectively [33]. Tubrication can effectively improve the wear resistance of materials, which presents good inspiration for our later work.

### 3.4. Worn Track Morphology and Wear Mechanism

Under the low load of 5 N, the corresponding SEM microscope morphology of the worn tracks surface in the FG and CG samples were displayed in Figure 10. Adhesive wear occurred when the FG and CG sample and the ball pressed against each other. Under the sliding speed of 1 mm/s, due to high hardness, the worn track surface of FG sample is mainly composed of some fine and uniform grooves, while the worn track surface of CG has a small amount of debris accumulation in addition to some grooves. The grooves are mainly caused by slight ploughing during friction experiments. When the sliding speed is 5 mm/s, the track surface of the FG sample is still mainly grooving due to its high hardness, while the track surface of the CG sample has a large amount of debris accumulation and delamination due to its low hardness. When the sliding speed is further increased to 10 mm/s, a large number of debris and glaze layer appear on the worn surface of FG and CG samples, in addition to the grooves. There are some micro pits and smooth regions on the worn surface. The EDS analysis of the marked position for Figure 10 and Figure 11 are listed in Table 2. A and B are in the glaze layer and smooth part of the track of the CG sample under the load of 5 N and the speed of 5 mm/s, respectively. The EDS analysis for A in the glaze layer of Figure 10d found that there is some O element, so the oxidation friction of the CG sample occurred in this condition. The oxide layer has a slight fracture. The C and D are in the FG and CG sample glaze layers, respectively, under the load of 5 N and speed of 10 mm/s. The quantitative analysis of elements at C and D indicated that the content of element O had increased. With the increase of sliding speed, the friction heat will make the surface temperature increase, which will lead to surface thermal softening and oxidation [34]. Damage or overload increase due to stress, speed, or temperature of the sliding pair and adhesion may lead to wear. Therefore, the wear mechanism of the two samples was mainly adhesive wear and oxidative wear. Through a careful analysis of the SEM images, it is found that CG samples are more likely to form adhesive debris, the worn surface is more likely to form glaze layers, and the oxide layer of the CG sample formed during oxidation friction is prone to fracture, which leads to an increase in specific wear rate and a decrease in wear resistance. This also corresponds to the track worn track section profile curve. The curve of CG samples has larger curve fluctuation and deeper depression in the middle, which is caused by many debris and glaze layers [35]. Therefore, under low load conditions, when the sliding speed is low, there are many wear particles and grooves. These features demonstrate that the primary wear mechanism of FG and CG samples is adhesive wear. With the increase in sliding speed, the main wear mechanisms of both samples are adhesive wear and oxidative wear. However, the glaze layer of the CG sample has many cracks, and the oxide layer formed on the CG sample surface is very easily broken, so the specific wear rate and friction coefficient of the CG sample is much higher than that of the FG sample.

Figure 11 shows the worn surface SEM microscope morphology of the FG and CG samples under the high normal load of 30 N. The width of worn tracks is much wider than that of the samples under the low load of 5 N. This is because the worn surface automatically accumulates around and increases the area of the worn track surface [36]. The average friction coefficient of the FG and CG samples shows that the friction coefficient of both samples is much lower under a high load than that under the low load. As can be seen from Figure 11a,b, when the sliding speed is 1 mm/s, a little worn debris appeared on the surface of the two samples, in addition to the grooves. CG sample has relatively more debris on its surface, so the specific wear rate is much higher than that of FG sample. When the sliding speed increases to 5 mm/s, both FG and CG samples tend to form a glaze layer, resulting in a large amount of debris accumulation. The E and F are in the glaze layer of FG and CG sample, respectively, under the load of 30 N and the speed of 5 mm/s. The analysis of element content in Table 2 shows that both E and F contain a large amount of element O. Therefore, oxidation friction caused by friction heat occurred as the sliding speed reached 5 mm/s, forming the oxide layers on the sample surface. The oxidized layer can be used as the lubricating layer of the alloy and interface, which can effectively reduce the specific wear rate [37,38]. However, the oxide layers formed on CG sample surface are fractured. Thus, on the contrary, the specific wear rate increases, and the wear resistance of the CG sample decreases. Fatigue and formation of cracks in surface regions due to tribological stress cycles result in the separation of material. Adsorbed layers and oxide films on the contacting surfaces may be broken through due to elastic and plastic deformation [39,40]. When the sliding speed is 10 mm/s, the surfaces of both samples seriously wear, with a large amount of debris accumulation and serious delaminating on the worn surface. Chemical reaction products are formed due to chemical interaction. At this time, the main wear mechanisms of the FG and CG samples are adhesive wear and oxidation wear.

Through careful comparison, as shown in the yellow square box in Figure 11f, it is found that the worn surface of CG sample forms an oxide layer due to friction heat at high sliding speed, but the oxide layer cracks under long-term load, and the worn surface undergoes severe plastic deformation. High local pressure between contacting asperities results in plastic deformation, adhesion, and consequently the formation of junctions locally [41]. The G and H are in the oxide layer of FG and CG samples, respectively, under the load of 30 N and the speed of 10 mm/s. However, because of high hardness, the surface of FG samples also oxidizes and forms oxide layers due to friction heat at high sliding speed and high normal load, and the degree of rupture is small. FG samples show better wear resistance compared with CG samples. This is also consistent with the previous friction coefficient data and specific wear rate data. The friction coefficient of FG sample is small, and the specific wear rate is low. Under high normal load, with the increase of sliding speed, the main wear mechanisms of the two samples are adhesive wear, oxidative wear, and plastic deformation. Compared with low loading, the phenomena of wear debris, glaze layers, and oxide layers occurred earlier. This indicates that both samples are more prone to oxidation friction under the high normal load. The oxide layer acts as a lubricating layer, resulting in a reduced coefficient of friction on the worn surface. Therefore, in the friction coefficient curve, the friction coefficient of both samples under high normal load conditions is smaller than that of under low normal load. However, the formation of the oxide layer is accompanied by the occurrence of oxidative wear, which leads to the accumulation of wear volume and the increase of specific wear rate. At this point, the oxide layer formed on the FG sample is more resistant to rupture. Therefore, the FG sample shows good wear resistance.

## 4. Conclusions

In the present study, the tribological properties of different grain sizes CoCrFeMnNi HEAs were systematically analyzed based on sliding experiments and worn surface morphology characterization under different sliding speeds and normal loads. The results can be summarized as follows:(1)Under the normal low load and low sliding speed, the main wear mechanism of FG and CG samples is adhesive wear. With the increase in sliding speed, the wear mechanisms are adhesive wear and oxidation wear. However, the material of CG is easy to soften under the action of friction heat during high sliding speed, and the oxide layer formed is easy to fracture.(2)Under the normal high load, the wear mechanisms of FG and CG samples are mainly adhesive wear, oxidation wear, and plastic deformation with the increase of sliding speed. The oxidation layer formed on CG sample has many cracks, and the wear surface of CG sample also has plastic deformation, resulting in an increase in friction coefficient and specific wear rate and a decrease in wear resistance. Therefore, under high load and low load, FG sample has good wear resistance due to its high surface strength.(3)Both sliding speed and normal load will affect the friction and wear behavior of the different grain size CoCrFeMnNi HEAs. Increasing the sliding speed will increase the friction coefficient and specific wear rate of the alloy. Thus, increase the normal load, reduce the friction coefficient, and decrease the specific wear rate of the alloy. This is because the oxide layer formed on the worn surface under high load plays a role in lubrication and reduces the friction coefficient and specific wear rate.

## Figures and Tables

**Figure 1 materials-16-01714-f001:**
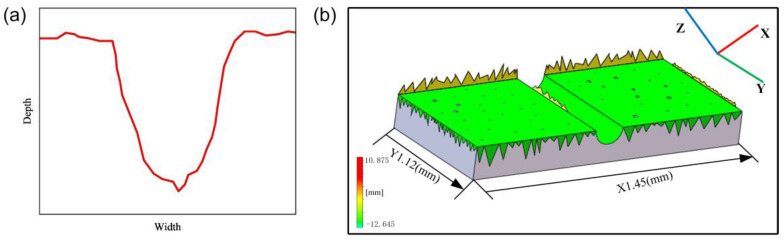
(**a**) Schematic diagram of the typical wear track profile and (**b**) 3D-profile image of a wear track.

**Figure 2 materials-16-01714-f002:**
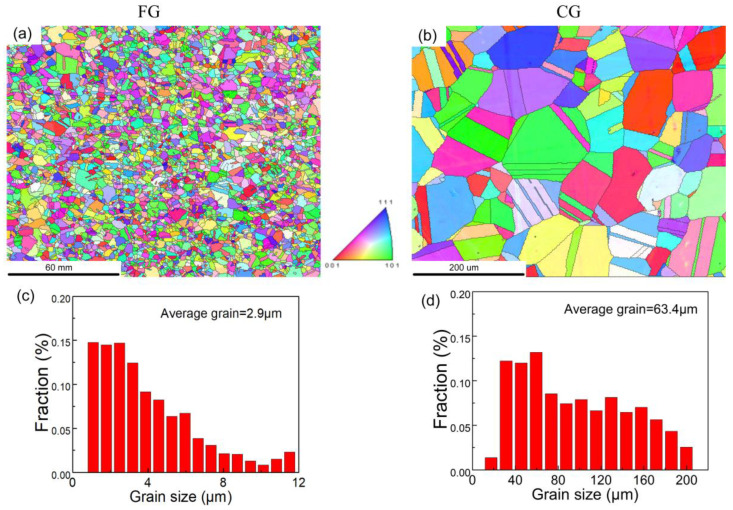
The microstructure of CoCrFeMnNi alloy after HEA treatment, (**a**) IPF diagram of FG, (**b**) IPF diagram of CG, (**c**) grain size of FG, and (**d**) grain size of CG.

**Figure 3 materials-16-01714-f003:**
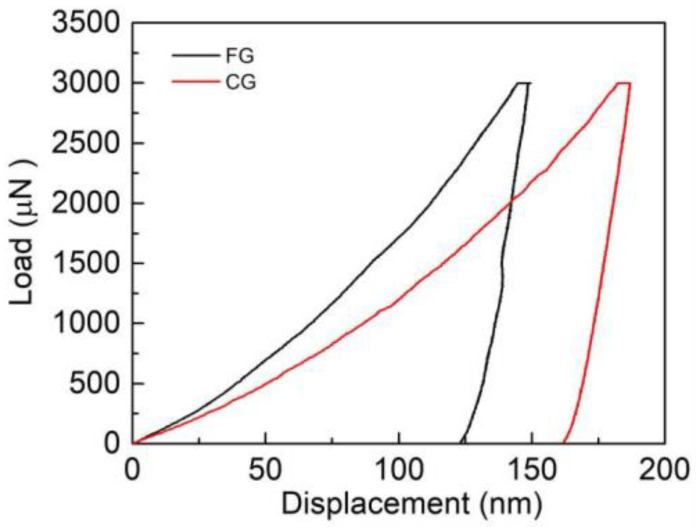
Nanoindentation load-displacement curves of FG and CG HEA sample.

**Figure 4 materials-16-01714-f004:**
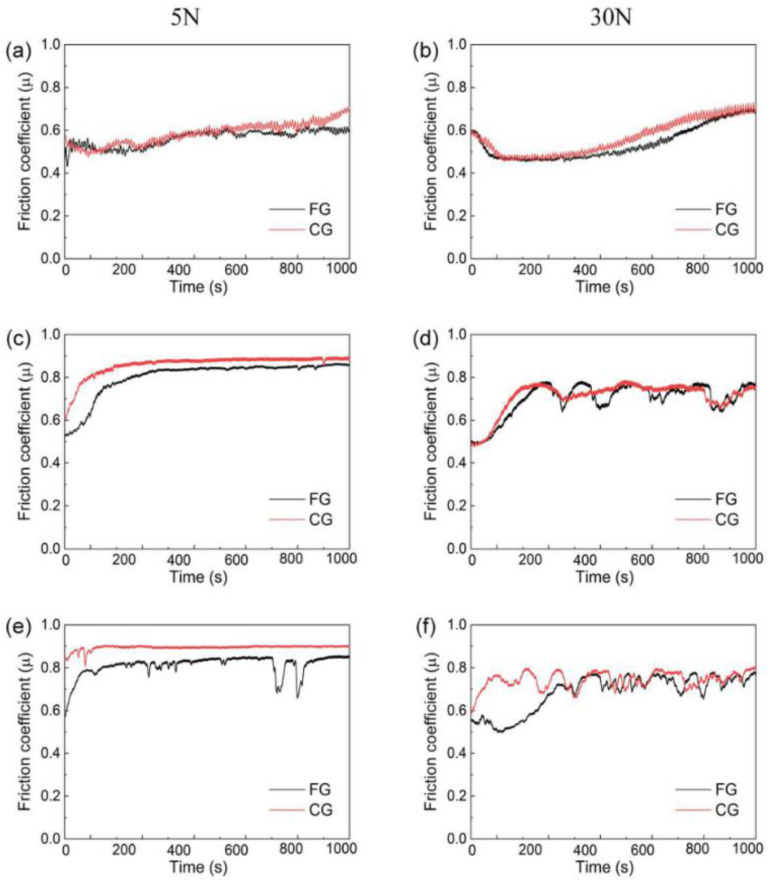
Friction coefficient curves of FG and CG HEA sample under different normal loads and speed, (**a**) 5 N-1 mm/s, (**b**) 30 N-1 mm/s, (**c**) 5 N-5 mm/s, (**d**) 30 N-5 mm/s, (**e**) 5 N-10 mm/s and (**f**) 30 N-10 mm/s.

**Figure 5 materials-16-01714-f005:**
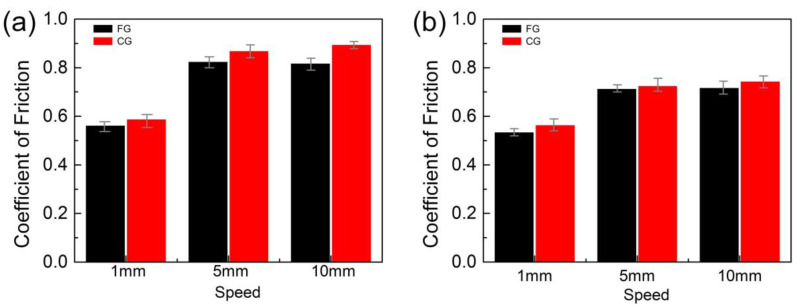
The average friction coefficient of the FG and CG sample at different loads, (**a**) 5 N and (**b**) 30 N.

**Figure 6 materials-16-01714-f006:**
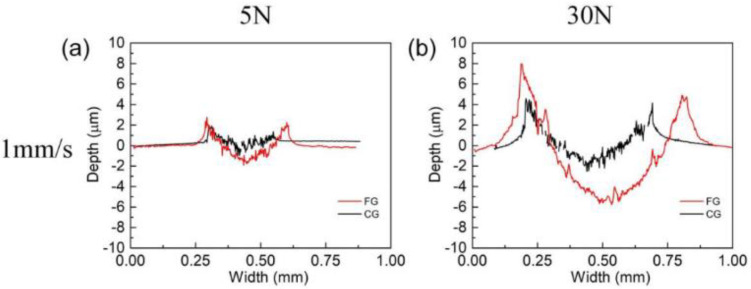
FG and CG HEA in different friction conditions of the worn track section profile curve, (**a**) 5 N−1 mm/s, (**b**) 30 N−1 mm/s, (**c**) 5 N−5 mm/s, (**d**) 30 N−5 mm/s, (**e**) 5 N−10 mm/s and (**f**) 30 N−10 mm/s.

**Figure 7 materials-16-01714-f007:**
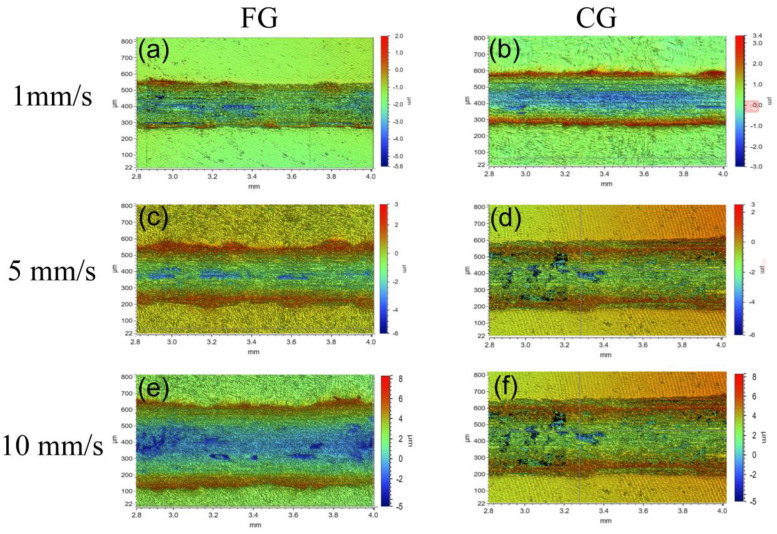
3D microscope morphology of track under normal load of 5 N, (**a**) FG−1 mm/s, (**b**) CG−1 mm/s, (**c**) FG−5 mm/s, (**d**) CG−5 mm/s, (**e**) FG−10 mm/s, and (**f**) CG−10 mm/s.

**Figure 8 materials-16-01714-f008:**
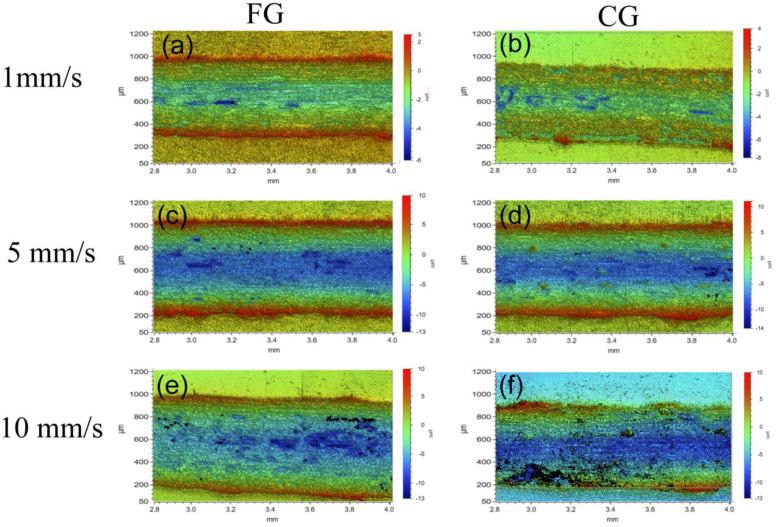
Three-dimensional microscope morphology of track under normal load of 30 N, (**a**) FG−1 mm/s, (**b**) CG−1 mm/s, (**c**) FG−5 mm/s, (**d**) CG−5 mm/s, (**e**) FG−10 mm/s, and (**f**) CG−10 mm/s.

**Figure 9 materials-16-01714-f009:**
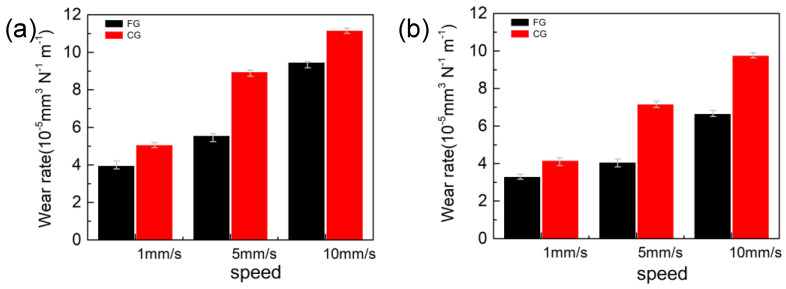
The average specific wear rate of the FG and CG sample at different loads, (**a**) 5 N and (**b**) 30 N.

**Figure 10 materials-16-01714-f010:**
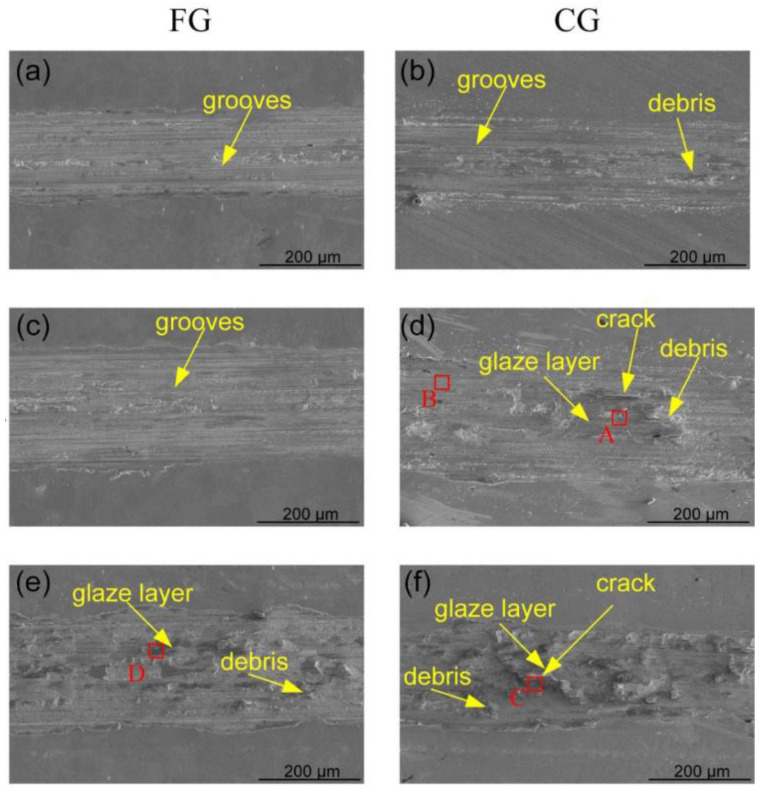
SEM microscope morphology showing the worn tracks of FG and CG samples under the normal load of 5 N and different sliding speeds, (**a**) FG−1 mm/s, (**b**) CG−1 mm/s, (**c**) FG−5 mm/s, (**d**) CG−5 mm/s, (**e**) FG−10 mm/s, and (**f**) CG−10 mm/s.

**Figure 11 materials-16-01714-f011:**
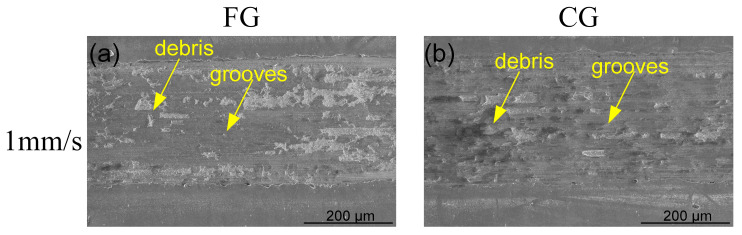
SEM microscope morphology showing the worn tracks of FG and CG samples under the normal load of 30 N and different sliding speeds, (**a**) FG−1 mm/s, (**b**) CG−1 mm/s, (**c**) FG−5 mm/s, (**d**) CG−5 mm/s, (**e**) FG−10 mm/s, and (**f**) CG−10 mm/s.

**Table 1 materials-16-01714-t001:** Compositions (at. %) for the CoCrFeMnNi HEA.

Element	Cr	Mn	Fe	Co	Ni
Content	19.58%	19.27%	20.05%	20.99%	20.11%

**Table 2 materials-16-01714-t002:** Element content in the marked position of samples morphology (at. %).

Position	Co	Cr	Fe	Mn	Ni	O
A	19.4	21.1	19.8	17.7	18.5	3.5
B	19.6	21.3	20.5	19.2	19.1	0.3
C	16.9	17.4	15.6	11.4	17.2	21.5
D	10.3	13.7	16.5	9.8	16.3	33.4
E	12	12.1	11.6	11.1	10.6	42.6
F	7.9	13.7	9.8	12.4	6.1	50.1
G	5.1	11.3	10.5	7.8	5.6	59.7
H	8.8	11.9	8.7	4.5	2.7	63.4

## Data Availability

No new data were created or analyzed in this study. Data sharing is not applicable to this article.

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
