# Peer review of "Effect of Grain Size on the Tribological Behavior of CoCrFeMnNi High Entropy Alloy"

_materials, 2023, doi:10.3390/ma16041714_

Round 1

Reviewer 1 Report

Although presenting interesting results, there are many grammatical mistakes in the text which make it impossible to understand. I urge the authors to correct the text before resubmitting this manuscript. 

Reviewer 2 Report

I suggest to the authors check the spelling of the whole paper. For example,

81 row: additional a,

133 row: EBSD is electron backscatter diffraction,

193: frication.

Only few questions came up:

1.     How was check the homogeneity of the HEAs samples

2.     Figure 2, what was the stepsize during the measurement? Have been used any cleaning processes on the EBSD maps? How was determined the grain size? What was the misorientation angle?

3.     The authors showed the grain boundary in Fig 2a, why did not show it in Fig 2b too?

Reviewer 3 Report

Linie 33-34

For each of the listed properties, appropriate references should be added.

Line 51:

Haw looks the final stoichiometry after doping of Si? (The doping of Si into Al0.2Co1.5CrFeNi1.5Ti0.5)

Line 63

Delete one of the dot “.[21]."

Linie 90

The composition was shown in Table 1. What does it mean “position” in Table1? The description of the position from A to H should be added.

Line 139

I’m confused. It should be add for which sample the microstructure was presented in Figure 2?  

Line 145

The grain size of the sample was quantitatively obtained by using the analysis software OIM in EBSD. How the program calculate the grain size? Is it some kind of average?

Reviewer 4 Report

Good afternoon, respected editor of the journal and colleagues!

I bring to your attention my opinion about the article "Effect of grain size on the tribological behavior of 2 CoCrFeMnNi high entropy alloy". 

In this paper was investigated effect and mechanism of grain sizes on the tribological properties of HEAs. 

After reading the article, a number of questions arose: 

1. How do you explain unsable behaviour of friction coefficient of FG condition? (Fig. 4)

2. On fig. 1, authors need to add a triangle of misorientations

Technical comments:

- in figs. 1-4, 6-8 authors used to small font

- in fig. 11 to small resolurion

Round 2

Reviewer 1 Report

The text has been corrected; given the comments and reviews of other reviewers and the additional modifications made by the authors, I think the manuscript is now fit for publication.